# Joint Learning for Visual Reconstruction from the Brain Activity: Hierarchical Representation of Image Perception with EEG-Vision Transformer

**Ali Akbari**
Department of Electrical Engineering
Sharif University of Technology
Tehran, Iran
ali.akbari@ee.sharif.edu

**Kosar Sanjar**
Graduate of Biomedical Engineering
University of Tehran ECE Department
Tehran, Iran
kosar.sanjar@gmail.com

**Muhammad Yousefnezhad**
Research Associate at University of Alberta
Departments of Computing Science and Psychiatry
Alberta, Canada
myousefnezhad@ualberta.ca

**Maryam S. Mirian**
Research Associate at UBC
Department of medicine
British Columbia, Vancouver, Canada
maryam.mirian@ubc.ca

**Emad Arasteh**
PhD candidate at the University Medical Center Utrecht
Utrecht, the Netherlands
e.arastehemamzadehhashemi@umcutrecht.nl

## Abstract

Reconstructing visual stimuli from brain activity is a challenging problem, particularly when using EEG data, which is more affordable and accessible than fMRI, though noisier and with lower spatial resolution. In this paper, we present Hierarchical-ViT, a novel framework designed to improve the quality and precision of EEG-based image reconstruction by integrating hierarchical visual feature extraction, vision transformer-based EEG (EEG-ViT) processing, and CLIP-based joint learning. Inspired by the hierarchical nature of the human visual system, our model progressively captures complex visual features—such as edges, textures, and shapes—through a multi-stage processing approach. These features align with EEG signals processed by the EEG-ViT model, allowing for the creation of a shared latent space that enhances contrastive learning. A StyleGAN is then employed to generate high-resolution images from these aligned representations. We evaluated our method on two benchmark datasets, EEGCVPR40 and ThoughtViz, achieving superior results compared to existing approaches in terms of Inception Score (IS), Kernel Inception Distance (KID), and Fréchet Inception Distance (FID) for EEGCVPR, and IS and KID for the ThoughtViz dataset. Through an ablation study, we underscored the feasibility of hierarchical feature extraction, while multivariate analysis of variance (MANOVA) test confirmed the distinctiveness of the learned feature spaces. In conclusion, our results show the feasibility and uniqueness of using hierarchical filtering of perceived images combined with EEG-ViT-based features to improve brain decoding from EEG data.

# 1 Introduction

Visual reconstruction is a critical area of research in both neuroscience and machine learning (ML), as it provides insights into how the brain processes and represents perceptible information [Takagi and Nishimoto, a]. In this regard, understanding the neural correlates of visual perception is vital for decoding brain activity, which has implications for both cognitive science and clinical applications [Pollen]. Traditionally, visual reconstruction methods have heavily relied on functional magnetic resonance imaging (fMRI) data, which allows for high spatial resolution images of brain activity [Rakhimberdina et al.]. However, fMRI-based approaches come with limitations, including high costs, limited accessibility, and (often) low temporal resolution, making them less practical for continuous monitoring or real-time applications [Glover, Wilson et al.]. To tackle the challenges mentioned above, there is a growing interest in EEG-based methods of visual reconstruction, as EEG offers a balance between temporal resolution and cost-effectiveness [Wilson et al.].

The field of image reconstruction from brain activity has recently advanced noticeably with the adoption of sophisticated generative models, including generative adversarial networks (GANs) and diffusion models with the Stable Diffusion [Ozcelik and VanRullen] ability to generate high-resolution images, showcasing the power of large pre-trained models [Takagi and Nishimoto, b] and offering new insights into visual processing. Moreover, recent advancements in AI and machine learning have opened possibilities for effective image reconstruction from EEG signals, addressing the need for cost-effective and efficient approaches in neuroimaging [Li et al., Guenther et al.]. Joint learning is one of these recent techniques that enables continuous data of EEG to be paired with other modalities like perceived images. Such pairing can help to synthesize new aspects of EEG-Image dynamics for more effective brain decoding [Song et al., Xu et al.]. The CLIP (Contrastive Language-Image Pretraining) model [Radford et al.] is a joint learning framework where a single model learns visual and language representations by predicting which caption matches which image. By extending this capability to EEG data, the model learns shared representations across different modalities, enhancing the potential for accurate visual reconstructions from EEG signals [Singh et al., a].

Regardless of their feasibility in aligning EEG data with perceived images, multimodal learning models like CLIP cannot provide a measure of a biologically plausible representation of decoded brain activity. This may be one important reason that most of the existing models for generating images from EEG data have primarily excelled in distinguishing between different classes of perceived images, rather than reconstructing the actual visualized image with high accuracy [Kavasidis et al., Jiang et al., a, Khare et al., Spampinato et al.]. On the other hand, several researchers have demonstrated the advantages of biologically inspired computational modeling for both advanced deep learning models' performance as well as more feasible biologically-inspired tools [Kim et al., Luppi et al., Collins and Shenhav]. Over the last two decades, several models for decoding visual perception mechanisms have been proposed that support the concept of hierarchical image processing in the brain, where different layers (e.g., V1, V2, V3, V4 in the visual cortex) process different aspects of visual stimuli. [Pohl et al., Bracci et al., D'Souza et al.].

In this paper, we propose a model (called here "Hierarchical-ViT") combining the vision transformer (ViT), hierarchical visual feature extraction, and contrastive learning to improve visual reconstruction from EEG signals. We utilized the EEG-ViT [Yang] model for EEG feature extraction, leveraging their self-attention mechanisms to capture complex temporal and spatial patterns in brain activity. These EEG features are integrated with hierarchical visual features, inspired by the human visual system's layered processing of visual stimuli. The combined EEG and visual features are aligned in a shared latent space using the CLIP framework, enhancing the model's ability to accurately reconstruct images from EEG data. Finally, a StyleGAN [Karras et al.] model is employed for high-resolution image generation, allowing for greater control and realism in the reconstructed visuals.

The paper is structured as follows: The Related works section reviews existing approaches and models for image reconstruction from brain activity. Then, the method section details our proposed approach. The experiment and results section presents the evaluation of the proposed method's performance. The discussion section interprets the results and explores their implications for future research. Finally, the conclusion summarizes the key contributions and potential directions for further development in EEG-based visual reconstruction.

## 2 Related works

Recent improvements in generative AI have encouraged researchers to develop new encode-decoder frameworks for image reconstruction by VAE, GAN, or latent diffusion models [Huang et al., a, Gong et al., Ozcelik and VanRullen]. Although the diffusion and VAE models have great advantages in terms of stability and versatility, GAN models are well-known for their ability to generate realistic images [Peng]. This can be helpful, especially in the endeavor of generating natural image generation by brain decoding where metrics like IS, KID, and FID are the major criteria of models' performance.

Researchers have developed attention-based GAN architectures that can reconstruct complex natural object images from EEG data, outperforming traditional cross-modality encoder-decoder networks [Habashi et al.]. These models often incorporate additional components such as perceptual loss and auxiliary classifiers to improve the quality and relevance of the generated images [Mishra and Bhavsar]. Other approaches have utilized contrastive learning methods to extract features from EEG signals, which are then used to condition GANs for image synthesis. These techniques have shown promise in reconstructing various types of visual stimuli, including objects, digits, and characters, even when working with small-scale EEG datasets [Hartmann et al.].

While GAN-based image reconstruction methods applied to EEG data have shown promising advancements, they still face challenges in achieving the same level of quality as similar techniques applied to fMRI data, particularly in terms of IS and FID [Yang and Modesitt]. Therefore, there is still room for improvement to elevate the performance of EEG data decoding for natural image reconstruction.

## 3 Method

We considered two already-established AI and cognitive science-known facts to enhance image reconstruction from EEG signals:

1. The self-attention mechanism of transformer models generally excel LSTM methods in terms of long-term dependencies and flexible context modelings. Moreover, in the case of EEG data analysis, Vision transformers (ViT) can extract more feasible spatial-temporal features compared to regular attention-based models [Yang and Modesitt].

2. The human visual system processes information through a hierarchical structure, where different specialized brain regions manage progressively more complex aspects of visual stimuli [Lerner et al.].

Based on these two facts, we propose two modifications to existing GAN-based image reconstruction models:

1. An EEG-ViT [Dosovitskiy et al.] for feature extraction from EEG instead of LSTM and CNN models to uncover longer-term spatial-temporal dynamics of the brain.

2. Hierarchical feature extraction from images with joint space learning to improve the biologically plausible signal processing compared to earlier methods.

These modifications form the foundation of our proposed image reconstruction framework, as illustrated in Fig. 1. While classification accuracy remains important, metrics such as IS, KID, and FID are more reflective of the quality of the reconstructed images. In this paper, we hypothesize that leveraging the EEG-ViT model alongside biologically inspired image feature extraction will enhance these three metrics. Accordingly, our goal is to achieve a joint learned representation of EEG and image features to improve the quality of reconstructed images. All experiments and models were trained on a server equipped with one V100 GPU card and 200 GB of RAM. The entire analysis took approximately 100 hours.

### 3.1 Notations

We refer to the feature vectors extracted from EEG and image as psi in Fig. 1. The EEG signal is denoted by $X$ and the image as $Y$. The EEG signal is in space $X \in \mathbb{R}^{N \times C \times T}$, where $N$ is the number of EEG trials, $C$ is the number of channels, and $T$ is the number of time points. The

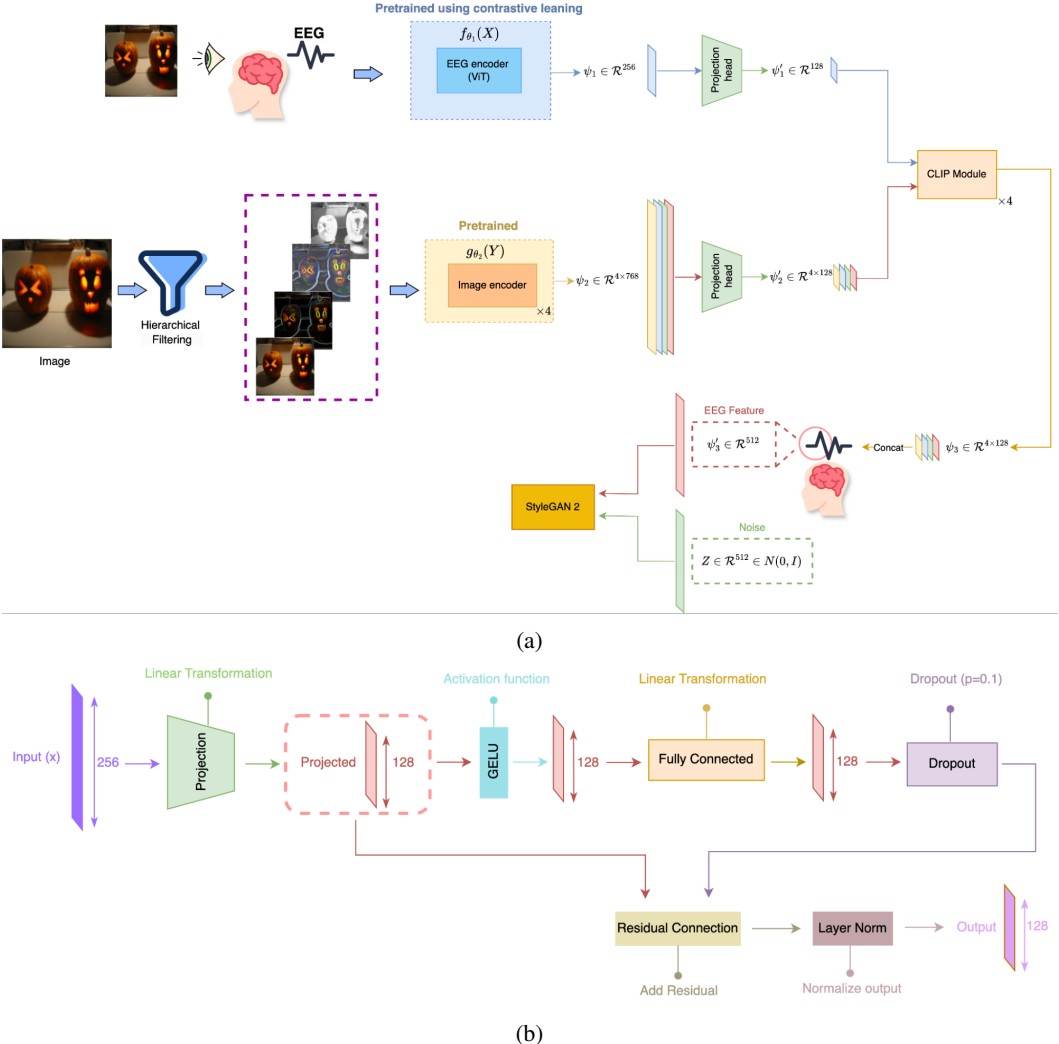

(a)

(b)

Figure 1: Hierarchical-ViT scheme. (a) Overview of the proposed framework for (training phase of) the EEG-based visual reconstruction using hierarchical feature extraction and contrastive learning. (b) The detailed structures of the "Projection head" are drawn in the general scheme. Detailed structures of the other components are drawn in the subsequent subsections through Fig. 2 and appendix A.

corresponding labels are marked as $L$. The EEG feature extractor is shown by $f_{\theta_1}(X)$ and the image feature extractor as $g_{\theta_2(X)}$, where $\theta_i$ is the learned weight during the training. The projection head for $g_\theta$ and $f_\theta$ are $g_\gamma$ and $f_\gamma$. We define the problem as follows: Given a data set of samples of $\{X, Y, L\}$, we want to train a deep neural network pipeline to reconstruct $Y$, given $X$ and $L$.

## 3.2 Transformer-based feature extraction from EEG data

Transformer models offer significant advantages over LSTM and CNN-based approaches for EEG feature extraction [Hu et al.]. By leveraging self-attention, they can capture long-range dependencies more effectively, improving the understanding of complex temporal patterns in brain activity [Siddhad et al.]. In this paper, we selected EEG-ViT models for their unique capabilities to effectively and simultaneously capture both the spatial relationships between EEG channels and the temporal dynamics of brain activity [Yang and Modesitt, Patel et al.].

During pre-training, the triplet margin loss is used, while Cross Entropy Loss is applied in the CLIP setting to maximize the similarity between image and EEG pairs. We use the triplet loss for feature learning with semi-hard triplets. We utilized a Vision transformer (ViT)-based model, EEG-ViT, for

EEG feature extraction. The Transformer treats the EEG data as image-like inputs, enabling it to model spatial-temporal relationships more effectively.

The EEG encoder $f_{\theta_1}(X)$ maps the EEG signals into a feature space $\psi_1 \in \mathbb{R}^{N \times 256}$. This encoded feature representation captures both local and global patterns from the input EEG signals. We apply a projection head to transform these features into a lower-dimensional latent space $\psi_1' \in \mathbb{R}^{N \times 128}$. To optimize this process, we pre-train the EEG encoder using contrastive loss. The triplet margin loss function is employed to align the EEG features with their corresponding visual representations. The loss function is given by:

$$\theta = \arg\min_{\theta} \mathbb{E}\left[||f_\theta(X^a) - f_\theta(X^p)||_2^2 - ||f_\theta(X^a) - f_\theta(X^n)||_2^2 + \delta\right]$$

where $X^a$, $X^p$, and $X^n$ represent anchor, positive, and negative samples, respectively. This helps ensure that the EEG features are closely aligned with the correct visual stimuli in the latent space.

### 3.3 Hierarchical visual features from the perceived natural images

Hierarchical models of the visual system are considered to correspond to four major feedforward V1-V4 layers by some neuroscientists [Riesenhuber and Poggio]. In the early processing stage of model area V1, boundaries and their orientations are detected, followed by a grouping process in model area V2. Contextual boundary patterns are also processed at a broader spatial level in model areas V2 and V4, allowing for sensitivity to contour curvature[Angelucci and Bressloff].

In this paper, we took advantage of simple feedforward hierarchical filtering of perceived images to use for joint learning with EEG features. The four hierarchical filters are as follows:

1. **V1 (Edge detection)**: The Sobel filter computes the gradient magnitude of an image to detect edges. The gradient magnitude is given by:

$$G = \sqrt{(S_x^2 + S_y^2)}$$

   Where:
   - $S_x$ and $S_y$ are the gradient in the x and y direction, respectively.

2. **V2 (Texture and contour detection)**: Local Binary Patterns (LPBs) and contour detection simulate V2's role in recognizing textures, contours, and boundary details, processing the information from the prior layers.

3. **V3 (Motion and color processing)**: The HSV [Smith, 1978] (Hue, Saturation, Value) color model separates color information into three channels. The saturation channel is extracted from the converted RGB color space to HSV color space and used as the main component extracted in V3.

   Detailed explanations are discussed in the Supplementary section. The outcomes of V4 were shown heuristically to be redundant in terms of the jointly learned feature space of the original image. Therefore, we used V1-V3 added to the original image for feature extraction from the image.

   We utilize a pre-trained ViT [Wu et al., Deng et al.], $g_{\theta_2}(Y)$ and fine-tune it on our image data set. Four models are trained on each one of the resulting image data sets after filtering in V1, V2, and V3 filters and the original images. The objective function is the contrastive loss. This gives us an embedding space $\psi \in \mathbb{R}^{256}$.

### 3.4 CLIP-based joint learning of image and EEG

Recent studies have explored innovative approaches to bridge the gap between EEG signals and visual representations. [Palazzo et al., a] employed a contrastive learning strategy with triplet loss to train an EEG encoder, aligning it with image features generated by a pre-trained image encoder. [Ye et al.] employed CLIP for joint representation learning by generating image representations via a GAN before training the EEG encoder using contrastive methods for image retrieval. On the other hand, Singh et al. [Singh et al., a] directly applied a pre-trained image encoder for EEG-based image retrieval tasks, streamlining the process and potentially improving efficiency.

We used the fine-tuned $g_{\theta 2}(Y)$ and pre-trained $f_\theta 1(X)$ in the CLIP module to align their embedding spaces, $\psi_1' \in \mathbb{R}^{128}$ and $\psi_2' \in \mathbb{R}^{128}$. We freeze $g_{\theta_2}$ and only allow $f_\theta$ and $g_\gamma$ and $f_\gamma$ to be updated in this process, due to the higher accuracy rate that $g_\theta$ possesses in our framework. This setup allows the EEG encoder to learn to better align its representation, resulting in better accuracy rates. Fig. 2 depicts the general graphical scheme of joint learning of EEG-Image feature spaces in our work.

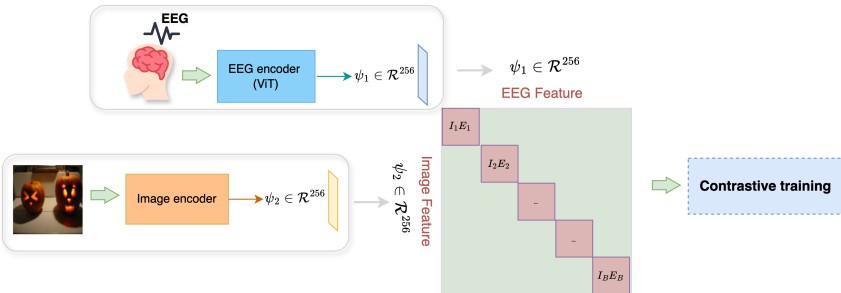

Figure 2: Joint learning of image and EEG features. The general scheme of CLIP-based joint learning in our work. This architecture allows us to align the representations of the EGG signals and the corresponding images. We trained four different CLIP modules, corresponding to V1, V2, V3 generated images and the original images. Each one of the CLIP modules is equipped with its corresponding pre-trained image encoder. Therefore, each EEG encoder learns a different embedding space.

We trained four different CLIP modules, corresponding to V1, V2, V3 generated images and the original images. Each one of the CLIP modules is equipped with its corresponding pre-trained image encoder. Therefore, each EEG encoder learns a different embedding space. The model was originated from [Shariatnia, 2021].

### 3.5 Image generation by Style-GAN model

StyleGAN includes several key features that improve image generation. It uses style blending with hybrid regularization and two random latent codes, giving users control over the style of the images for greater customization [Huang et al., b]. The model is capable of producing high-resolution images and managing complex tasks like face and landscape generation, reducing common issues like blurring and distortion seen in traditional GANs. Additionally, StyleGAN introduces stochastic variation by adding uncorrelated Gaussian noise to each layer of the network, which helps generate diverse and varied details. This allows the generated image to have some random variations in detail while maintaining overall structural consistency, increasing the diversity and realism of the image [Karras et al.].

We concatenated the resulting EEG features from the CLIP model with a normal vector $z \in \mathbb{R}^{512}$, resulting in a vector $\mathbb{R}^{1024}$. The model is trained to reconstruct the images, corresponding to the EEG feature vector.

### 3.6 Unique feature space and ablation study of hierarchical image features

While previous research has shown the biological plausibility of hierarchical feature extraction from perceived images [Horikawa and Kamitani, DiCarlo et al., Serre et al., Riesenhuber and Poggio], to the best of our knowledge, the importance of these features for improving brain decoding models has not been thoroughly explored. For this aim, we first evaluated the uniqueness of the feature space out of joint learning for V1-V3 extracted features compared to the feature space learned from the original image. We performed a MANOVA test on the generated feature spaces to evaluate the uniqueness of each feature space. In the next step, we applied an ablation study to evaluate the effect of these hierarchical features on the quality and precision of the generated images. An ablation study is a methodical technique in machine learning research used to analyze the influence of individual components or features on a model's performance. The process entails selectively removing or modifying certain elements of the model, retraining it, and then evaluating the impact of these alterations on its overall performance [Meyes et al.]. We compared the results of Hierarchical-ViT

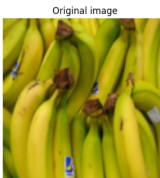 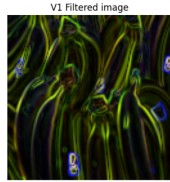 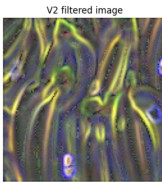 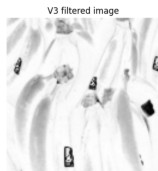

Original image      V1 Filtered image      V2 filtered image      V3 filtered image

Figure 3: The filtering effect on a sample image

with the CLIP-ViT model in which only features of the original image are utilized for joint learning with EEG features.

## 4 Experiment and results

The first part of this section discusses the two datasets. The second part discusses the outcomes of feature extraction. After that, hierarchical feature extraction is discussed. Joint learning, image synthesis, generated images, and ablation study results are the next subsequent presented parts of this section.

### 4.1 Datasets

**EEGCVPR** This dataset [Spampinato et al.] is a subset of ImageNet [Deng et al.], including data of 40 object classes, with 50 images per class, for a total of 2,000 images. We utilized the version of 5-95 Hz of the EEGCVPR dataset to include the biggest possible bandwidth of frequency components in the data. The recording protocol presented visual stimuli to users in a block-based setting, showing images of each class consecutively in a single sequence. Each EEG segment contains data from 128 channels, recorded for 0.5 seconds at a 1 kHz sampling rate. The resultant EEG signal consists of 440 time samples, after discarding the first and the last time samples. We used the original prep-processing proposed by the authors [Spampinato et al.].

**ThoughtViz** This dataset [Tirupattur et al.] includes 10 different categories of objects. The images were shown to the participants and were asked to imagine the image that was shown to them. The EEG signal consists of 14 channels. The sampling frequency of the device is 128 Hz. After pre-processing that was proposed by [Tirupattur et al.], each epoch has 32 time steps.

### 4.2 Jointly learned EEG-image features by CLIP

Feature extraction from images is performed using a Sobel filter for edge detection (V1), LBP for texture features (V2), and color intensity (V3). These images are fed into the 3 different CLIP models, allowing them to learn different weights and feature spaces. The original image is also fed into another clip model. In Fig. 3, one example of the effect of each V1-V3 filter on the original image is depicted.

The EEG-ViT architecture consists of 3 layers. The MLP dimension is set to 64 and the number of attention heads, and the attention dimension is set to 16. A dropout rate of 0.5 is applied consistently across models. To further reduce the risk of overfitting, FTsurrogate and smooth time masking are employed for EEG data augmentation, techniques that have been shown to enhance performance in BCI tasks involving EEG signals. We used Adam optimizer alongside cosine learning rate scheduler, $\phi$ of FTsurrogate equal to 1, and 0.5 probability of data augmentation for both of the data sets.

Fig. 4 depicts the nonlinearly mapped two-dimensional feature space from EEG-ViT by t-SNE [Maaten and Hinton] for the EEGCVPR dataset. The feature spaces of the ThoughtViz dataset are also depicted in the Supplementary section.

### 4.3 Generated images by StyleGAN

A group of images generated for each of the datasets by our model is shown in Fig. 5.

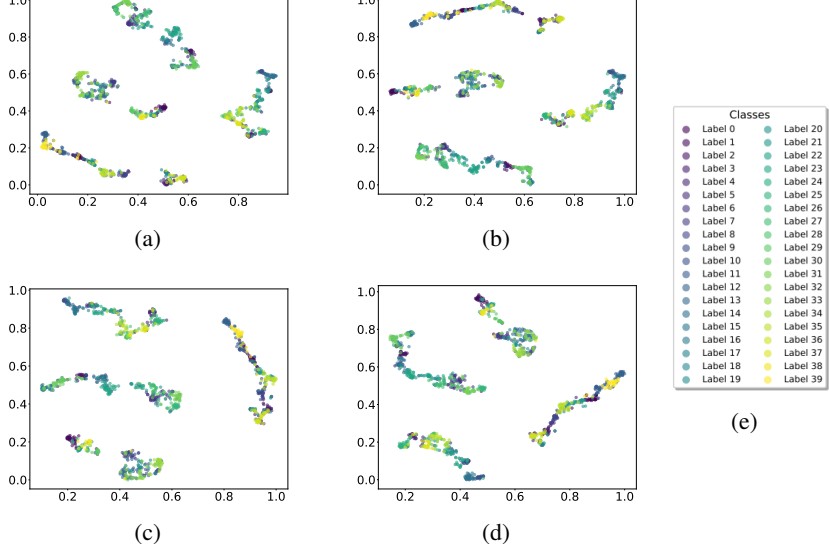

(a)  (b)  (e)

(c)  (d)

Figure 4: The result related to effects V1-V3 filtering on the feature space. The figures illustrate the first two dimensions of the t-SNE map [Maaten and Hinton] (horizontal as the first dimension). CLIP-based Jointly learned feature space of (a) original image, (b) V1-filtered image, (c) V2-filtered, and image (d) V3-filtered image.

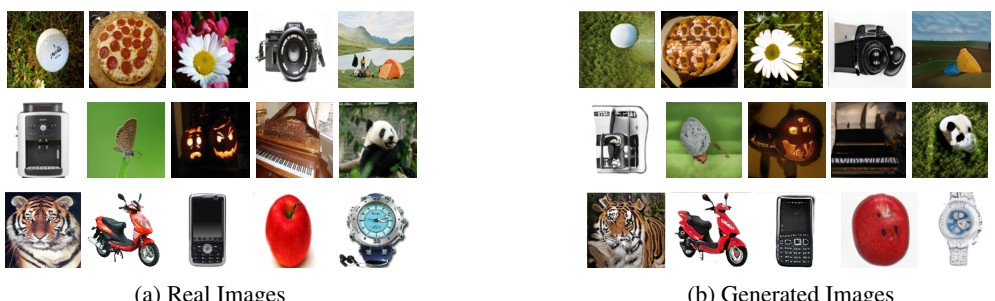

(a) Real Images  (b) Generated Images

Figure 5: Comparison of real perceived images and generated images by Hierarchichal-ViT model for EEGCVPR [Spampinato et al.] and ThoughtViz [Tirupattur et al.] datasets. The first two rows are related to EEGCVPR and final row are from ThoughtViz.

Table 1 summarizes the classification accuracy and quality of the generated images among the two data sets for our method and already established methods in the field of image generation from EEG. Our Hierarchical-ViT model, combined with StyleGAN, generates high-resolution images from EEG data for both the EEGCVPR40 and ThoughtViz datasets. The hierarchical extraction of visual features, aligned with EEG representations, improves the quality of generated images. Results show superior outcomes in IS, FID, and KID for the EEGCVPR dataset and IS and KID for ThoughtViz.

## 4.4  Comparison of feature spaces and ablation study

We compared the four feature spaces depicted in Fig. 4 by multivariate analysis of variance (MANOVA) [Tabachnick and Fidell]. The MANOVA test showed that the V1, V2, and V3 feature spaces differ from the feature space related to the original image significantly (Pillai's Trace = 2.9960, $F(384, 13437) = 26053.44$, $p < 1e-5$). Therefore, each of the V1-V3 feature spaces had unique feature representations and were not redundantly adding dependent variables to the Hierarchical-ViT model. The detailed outcomes of MANOVA statistics are mentioned in the Supplementary section. For the ablation study (as mentioned in section 3.6), we evaluated the outcomes of the IS, FID, and KID by using only a learned representation of the original image. As mentioned in Table 1, the IS, FID, and KID of our model is 12.17, 122.91, and 0.059, while for the same model,

these results degraded to 11.17, 126.88, and 0.062. Therefore, this ablation study also shows the feasibility of adding hierarchical feature extraction for more accurate decoding of visual processing from brain activity.

Table 1: Between-class discrimination by extracted EEG features and quality of the generated images. The discriminative metrics are calculated based on accuracy and k-means[Macqueen, 1967]. The k-means algorithm is applied to the features generated by our model. This table compares various approaches and loss functions for extracting generated images.

| Dataset | Reference | Discriminative Model | Classification | | Generative Model | Quality Metrics | | |
| --- | --- | --- | --- | --- | --- | --- | --- | --- |
| | | | Accuracy | K-Means | | IS ↑ | FID ↓ | KID ↓ |
| EEG CVPR | [Kavasidis et al.] | - | - | - | Brain2Image-VAE | 4.49 | - | - |
| | [Palazzo et al., b] | - | - | - | Brain2Image-GAN | 5.07 | - | - |
| | [Spampinato et al.] | LSTM Encoder | 0.829 | - | - | - | - | - |
| | Jiang et al. [a] | DML | 0.977 | - | - | - | - | - |
| | Zheng et al. | LSTM-CNN | 0.944 | - | Improved-SNGAN | 5.53 | - | - |
| | Jiang et al. [b] | BioLSTM | 0.991 | - | - | - | - | - |
| | Khare et al. | NeuroVision | 0.988 | - | - | 5.15 | - | - |
| | Singh et al. [a] | EEGLSTM | 0.983 | 0.96 | EEGStyleGAN-ADA | 10.82 | 174.13 | 0.065 |
| | | EEG-ViT (ours) | **0.72** | **0.70** | Hierarchical-ViT (ours) | **12.17** | **122.91** | **0.059** |
| ThoughtViz | [Tirupattur et al.] | ThoughtViz | 0.729 | - | ThoughtViz | 5.43 | - | - |
| | Mishra and Bhavsar | SiameseCNN | 0.741 | - | NeuroGAN | 6.02 | - | - |
| | Singh et al. [b] | EEG2Image | 0.55 | - | EEG2Image | 6.78 | - | - |
| | Singh et al. [a] | EEGLSTM | 0.741 | 0.72 | EEGStyleGAN-ADA | 9.23 | 109.49 | 0.039 |
| | | EEG-ViT (ours) | **0.85** | **0.84** | Hierarchical-ViT (ours) | **10.20** | 167.92 | **0.037** |

## 5    Discussion

This paper introduced a new approach for reconstructing images from EEG signals by combining transformer-based EEG feature extraction, hierarchical visual processing, and joint learning using the CLIP framework. The method improved the mapping of EEG signals to visual stimuli, leading to enhancements in the precision and quality of the generated images. On the EEGCVPR40 dataset, our model achieved an IS of 12.17 and a KID of 0.059, outperforming all the earlier established methods listed in Table 1. Similarly, with the ThoughtViz dataset, our model achieved an IS of 10.20 and an KID of 0.037, surpassing the other approaches. In the case of FID, our model outperformed all the other approaches by the value of 122.91 for the EEGCVPR dataset, while the EEGStyleGAN-ADA [Singh et al., a] exceeded our FID value (109.42 vs 167.92) for the ThoughtViz dataset. This shows the superiority of our model in 2 of 3 metrics to all other methods, with even better FID for a longer duration of the EEG dataset (EEGCVPR compared to ThoughtViz). These outcomes demonstrate the benefits of integrating hierarchical visual processing with transformer-based EEG feature extraction.

A key aspect of this approach is the use of hierarchical visual features, modeled after the layered structure of the human visual system. Results from the MANOVA test confirm that the features generated at each stage (V1, V2, V3) are unique and non-redundant, showing significant differences from the original image feature space (with p-value<1e-5). This highlights that feature spaces of hierarchical images are not redundant considering the original images' feature spaces. Moreover, the effectiveness of these hierarchical features was further confirmed by the mentioned ablation study showing lower quality images in terms of IS (11.17 vs. 12.17), FID (126.88 vs. 122.91), and KID (0.062 vs. 0.059).

One important limitation of our model is the lower classification accuracy for the EEGCVPR dataset compared to other methods (while our model reached the highest accuracy for ThoughtViz). It is worth mentioning that our model reached the highest image quality metrics on this dataset while attaining the lowest between-class discrimination. This can be a starting point for future works on the possible trade-off between class accuracy vs. precision of generated images in the brain-decoding field of research. Regardless of the promising results of this study, some challenges remain for the future steps. The complexity of transformer architectures can increase the risk of overfitting, especially with smaller datasets. While the hierarchical visual processing approach has proven to be beneficial, further work is needed to test its generalizability across different EEG datasets and a broader range of stimuli. Future studies could also explore alternative architectures or more efficient data augmentation methods to address the balance between model complexity and dataset size.

Moreover, new studies could also focus on better fine-tuning of Hierarchical-ViT model considering the length of EEG datasets in case of any relation between FID metric and the model's parameters.

## 6 Conclusion

This paper introduced a novel EEG-based image reconstruction method that integrates hierarchical feature extraction with transformer-based EEG processing and CLIP-inspired joint representation learning. By mimicking the human visual system's layered architecture, our approach effectively captured essential visual features like edges, textures, and contours, enhancing the alignment between EEG signals and visual representations. Our method outperformed existing techniques on datasets such as EEGCVPR40 and ThoughtViz, achieving superior Inception Score and Fréchet Inception Distance. Ablation studies confirmed the significance of hierarchical feature extraction in improving image quality and model robustness, while MANOVA tests validated the unique contributions of these features at various stages. Despite these promising results, challenges remain in generalizing to smaller, diverse datasets. Future research will focus on addressing model complexity and dataset scalability, alongside exploring efficient data augmentation strategies to enhance performance and mitigate overfitting.

In conclusion, the combination of hierarchical feature extraction with advanced EEG processing marks an advancement in EEG-to-image synthesis, while paving the way for more accurate brain decoding systems.

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

# A  Appendix: supplemental materials

## A.1  Hierarchical scheme

In this paper, we took advantage of simple feedforward hierarchical filtering of perceived images to use for joint learning with EEG features. We observed that heuristically, the simple four layers of

filtering without feedback work best on the limited benchmarking datasets. The four hierarchical filters are as follows:

- **V1 (Edge detection)**: Sobel filtering captures the role of V1 in detecting simple edges and orientation. The Sobel filter computes the gradient magnitude of an image to detect edges. The gradient magnitude is given by:

$$G = \sqrt{(S_x^2 + S_y^2)}$$

  where:
    - $S_x$ is the gradient in the x direction.
    - $S_y$ is the gradient in the y direction.

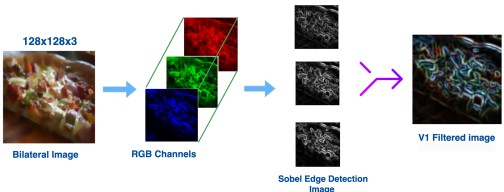

Figure A1: V1 filtering process

- **V2 (Texture and contour detection)**: Local Binary Patterns (LPBs) and contour detection simulate V2's role in recognizing textures, contours, and boundary details, processing the information from the prior layers. The LBP operator describes the local texture of an image by comparing each pixel with its neighbors. Contours represent the boundaries and edges in an image, being one of the most important items required to identify objects and shapes.

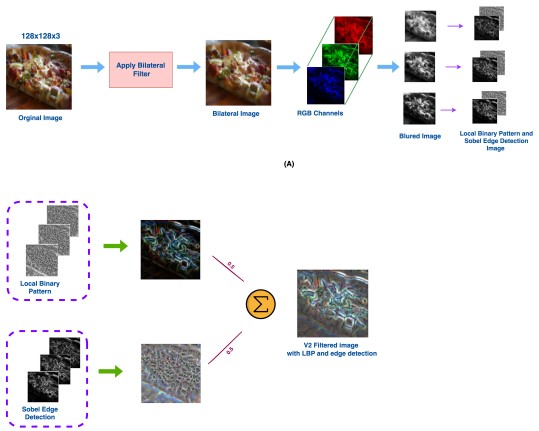

Figure A2: V2 filtering process

- **V3 (Color processing)**: Color saturation filtering reflects V3's role in processing dynamic information and complex color features, processing the information of the perceived sight. The HSV (Hue, Saturation, Value) color model separates color information into three channels. The saturation channel is extracted as:

$$S = \text{Saturation}(H, S, V)$$

  where:
    - $H$ is the Hue channel.
    - $S$ is the Saturation channel.
    - $V$ is the Value channel.

V4 (Curvature and shape detection): V4 is characterized as a mid-tier cortical area in the ventral visual pathway, positioned between earlier visual areas like V1/V2 and higher-level areas in inferotemporal cortex. Hough circles and shape detection simulate V4's involvement in higher-level shape and form recognition, which is crucial for object identification. The outcomes of V4 were shown heuristically to be redundant in terms of jointly learned feature space of the original image. Therefore, we used V1-V3 added to the original image for feature extraction from the image.

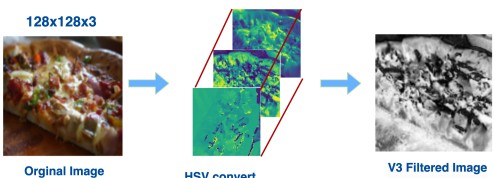

Figure A3: V3 filtering process

## A.2 MANOVA Analysis

The results of the MANOVA test can be seen in Table B1. The significance of the difference is presented as F Value and probabilities. All 4 tests yield high indication of significance. The Wilks' Lambda for the group is nearly zero. This indicates that nearly all the variability in the EEG features can be explained by group differences, suggesting very strong separation between the models. The Pillai's Trace is also high (2.9960), which confirms that a substantial amount of variance in the features is explained by the group effect. Hotelling-Lawley and Roy's Greatest Root are very high. This supports the idea that the groups are distinct.

Table B1: Multivariate Linear Model Results for EEG Features

| Test Statistic | Value | Num DF | Den DF | F Value | Pr > F |
|---|---|---|---|---|---|
| **Intercept** | | | | | |
| Wilks' Lambda | 0.0016 ↓ | 128 | 4477 | 22338.16 ↑ | 0.00001 ↓ |
| Pillai's Trace | 0.9984 ↑ | 128 | 4477 | 22338.16 ↑ | 0.00001 ↓ |
| Hotelling-Lawley | 638.6608 ↑ | 128 | 4477 | 22338.16 ↑ | 0.00001 ↓ |
| Roy's Root | 638.6608 ↑ | 128 | 4477 | 22338.16 ↑ | 0.00001 ↓ |
| **-V3 and original** | | | | | |
| Wilks' Lambda | 0.00001 ↓ | 384 | 13430.15 | 27048.62 ↑ | 0.00001 ↓ |
| Pillai's Trace | 2.9960 ↑ | 384 | 13437 | 26053.44 ↑ | 0.00001 ↓ |
| Hotelling-Lawley | 2404.6657 ↑ | 384 | 13078.52 | 28027.41 ↑ | 0.00001 ↓ |
| Roy's Root | 1073.1941 ↑ | 128 | 4479 | 37553.41 ↑ | 0.00001 ↓ |

## A.3 ThoughtViz embedding space

We also include the results of the embedding space for ThoughtViz dataset below.

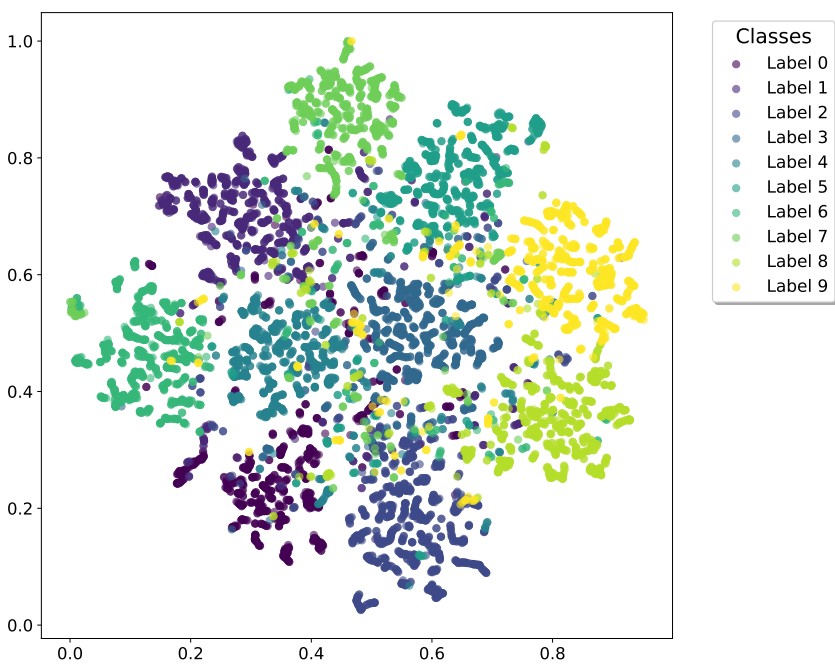

Figure A4: The t-SNE [Maaten and Hinton] plot for ThoughtViz dataset. The figure illustrates the first two dimensions of t-SNE map (horizontal as the first dimension).

