# OpenReview forum: "Joint Learning for Visual Reconstruction from the Brain Activity: Hierarchical Representation of Image Perception with EEG-Vision Transformer"
_NeurIPS.cc/2024/Workshop/UniReps — UniReps_

### Official Review · Reviewer_AX96 · 2024-09-29
**The topic is quite interesting and has great potential, but some information is missed and needs to be clarified.**

**Rating:** 6
**Confidence:** 4

**Review:**

The paper attempts to reconstruct visual images from EEG recordings of brain signals. Hierarchical visual image features are extracted using a feature extractor based on low-level visual functions. These extracted features are then fused with the brain signals using CLIP, followed by the application of StyleGAN to generate high-resolution images. This topic is quite interesting, as most studies typically use fMRI signals for visual image reconstruction. Given that EEG is more feasible and cost-effective, this research holds significant potential for brain-computer applications.

However, I have several questions that need to be addressed,

1. I prefer not to use the method described in the paper for extracting image features. While it’s acceptable to refer to features like edges, textures, and colors, using terms like V1, V2, V3, and V4 isn’t accurate. Strictly speaking, from a biological perspective, it’s not ideal to describe the process in that way. In the human visual system, RGB images are converted to LMS space and processed through different channels, such as frequency, color, and motion, allowing for parallel processing of information. Additionally, there are more complex computations involved, like divisive normalization.Therefore, I suggest that the authors revise the wording to avoid excessive references to biological vision and instead focus on pure computer vision/feature extraction concepts.

2. I hope the authors can provide the original images, the reconstructed images from EEG (your proposed method), and the reconstructed images from other methods. It would be helpful to present all these images together, rather than just the results as shown in Figure 5. Now, It’s hard to judge the reconstruction quality from EEG signals.

3. Please provide more detailed information about how you trained/fine-tuned StyleGAN, as well as details on the parameters related to contrastive learning. Currently, this information seems to be lacking.

4. Please include how you performed the preprocessing for EEG in the main text, as different preprocessing methods may lead to variations in the quality of the reconstructed images.

5. Please double-check the reference style, as most entries appear to be incorrect. One example here,

     [38] M. Riesenhuber and T. Poggio. Hierarchical models of object recognition in cortex. 2(11): 1019–1025. ISSN 1097-6256. doi:
     10.1038/14819.

     The journal name is missing.

6. In the appendix section A.2 (MANOVA Analysis), the referenced table number is missing (Table ??, on page 15).

7. In the appendix of section V3 (Motion and Color Processing), please remove "motion" since there is nothing related to motion in that section.

---

### Official Review · Reviewer_XbaQ · 2024-10-01
**Interesting work, some details missing**

**Rating:** 5
**Confidence:** 4

**Review:**

The paper presents a novel framework for EEG-based image reconstruction using a vision transformer-based approach, named "Hierarchical-ViT." It also utilizes CLIP-based joint learning and StyleGAN for high-resolution image generation. While the paper introduces novel contributions, some relevant literature is missing.

Specifically:

- Palazzo et al.'s Dataset Debate: The paper should reference the debate around Palazzo's dataset. This includes considerations on the limitations and challenges in using the dataset for EEG-based visual reconstruction and decoding tasks.

- The work titled "Training on the test set? An analysis of Spampinato et al." by Lin et al is not mentioned but is relevant to the field. This analysis discusses the potential pitfalls in training neural models on EEG data without proper generalization considerations, which is a crucial point when assessing the efficacy and validity of models like the proposed Hierarchical-ViT trained on this ImageNet EEG dataset.

- Some other relevant literature is missing

- The paper aims to reconstruct images from EEG data through a joint learning framework, but it’s not explicitly clear whether labels are used during inference. Typically, in such models, labels are used during training to align EEG features with visual features in a shared latent space, but they are not used during inference. However, due to the phrasing in the paper, this point might not be explicitly clarified, so it would be good to confirm with the authors or look for any specific section discussing inference directly to verify this assumption.

---

### Official Review · Reviewer_dbK2 · 2024-10-07
**The authors proposed a method for EEG-based reconstruction by integrating hierarchical visual features with CLIP joint learning. The proposed method most often achieves superior results compared to baselines.**

**Rating:** 6
**Confidence:** 3

**Review:**

The authors proposed a method for EEG-based reconstruction by integrating hierarchical visual features with CLIP joint learning. The proposed method most often achieves superior results compared to baselines.

Strengths
- The motivation is clear, addressing the challenge of using EEG data as opposed to fMRI.
- The results demonstrate that the proposed method performs well for both classification and reconstruction tasks.
- Ablation studies show that the hierarchical features encode distinct information, as evidenced by statistical tests.

Weaknesses
- While the authors report that MANOVA tests validate the unique contributions of these features, there is no clear evidence that hierarchical filtering improves reconstruction beyond encoding information at different levels. It would be helpful to include experiments comparing reconstruction performance with hierarchical filtering versus using normal image use.

Comments
- I am curious how this method would perform with fMRI data instead of EEG. Would the reconstructions be better? Including experiments that test this would help clarify whether the observed improvements stem from the method itself or from its adaptation to EEG data.

---

### Decision · Program_Chairs · 2024-10-10

**Decision:**

Accept

**Comment:**

In light of the positive reviewers' feedback and relevancy of the submission, we are pleased to accept this paper for presentation at UniReps 2024. We kindly ask the authors to incorporate the reviewers' suggestions and feedback in the final camera-ready version of the manuscript.